# Introduction to the potential of *Ferula ovina* in dental implant research due to estrogenic bioactive compounds and adhesive properties

Hoda Zare Mirakabad[1]*, M. Reza Khorramizadeh[2,3]*

**1** Department of Anatomy, Physiology and Pharmacology, University of Saskatchewan, Saskatoon, Canada, **2** Biosensor Research Center, Endocrinology and Metabolism Molecular-Cellular Sciences Institute, Tehran University of Medical Sciences (TUMS), Tehran, Iran, **3** Zebrafish Core Facility, Endocrinology and Metabolism Research Institute (EMRI), TUMS, Tehran, Iran

\* ho_za9@mail.um.ac.ir, hodazma313@gmail.com (HZM); khoramza@tums.ac.ir (MRK)

**Data Availability Statement:** All relevant data are within the manuscript and its Supporting Information files.

## Abstract

Recent developments in dental implant have heightened the urgent need to natural tissue adhesives estrogenic materials with ability of promoting the proliferation and osteoblastic differentiation in human dental pulp-derived stem cells, to provide better integration of tissue for dentistry. Up to now, far little attention has been paid to adhesives extract of the root of *Ferula* sp. which contains biomaterial compounds with estrogenic activities. Prior to undertaking the investigation, analysis of the extract of the root of *F. ovina* revealed a novel terpenoid, and we identified it as Fenoferin. So far, this paper has focused on Fenoferin compared to Ferutinin and root extract to determine if Fenoferin caused changes in craniofacial cartilage, bone (ceratohyal) and tooth mineralization. Following the purpose of study, we used zebrafish as a well-developed model system for studying bone development, so the developing zebrafish larvae were exposed to various concentration of compounds at 2dpf, and the histological analyses were performed at 6dpf. The result of the current study highlights the importance of *F. ovina* in studies related to dental regenerative medicine.

## 1. Introduction

Periodontal tissue consists of cementum, periodontal ligament (PDL) and alveolar bone. The loss of alveolar bone is responsible for changing the quantity of the alveolar ridge. Since formation of bone defects and the bone remodeling after tooth loss makes dental implant placement difficult or unfeasible depending on the size and location, reestablishment and maintenance of dimensions of the alveolar ridge are essential for a favorable implant [1]. The production of the reliable and predictable methods to stimulate bone regeneration in alveolar bone defects was the focus of studies on dental implant over the last few decades. Finding a way to restore the periodontal tissues and their function is the goal of periodontal regenerative techniques. The last studies suggested using of bioinspired implants combined with biomaterials and

**Funding:** The author(s) received no specific funding for this work.

**Competing interests:** The authors have declared that no competing interests exist.

cellular platforms for optimizing the benefits of complete restoration of human periodontium or hybrid interfaces against implants [2]. Raju *et al.* used an anatomically designed bioengineered complex cell sheet composed of PDL cells and osteoblast-like cells which connected PDL-like fibers to the tooth root and alveolar bone, functionally [3].

Estrogens are essential for normal skeletal maturation and proportions, accretion and maintenance of bone mineral density and mass, and control of the rate of bone turnover [4]. The estrogen and estrogen receptors (ERs) have important functions, not only in cortical and trabecular bone, but also in different bone cell types. Deficiency of estrogen could change the balance between bone resorption and bone formation, which could lead to osteoporosis and increase the risk of bone fracturing in menopause women [5]. Moreover, the balance between osteoclast and osteoblast is vital for a favorable dental implant [2]. The Woman Health Initiative showed that Hormone Replace Therapy (HRT) prevents the bone fracturing due to the osteoporosis induced by estrogen deficiency, but it could increase the risk of breast cancer, heart disease and stroke in women taking HRT [5].

Interestingly, phytoestrogens exert a protective effect against cardiovascular disease, menopausal symptoms (osteoporosis) and cancer [6, 7]. Ferutinin is a phytoestrogen with potential therapeutic agent in diseases induced by estrogen deficiency, which could reduce the side effects produced by usual estrogen therapy because of the cross-talk between the estrogenic and ionophoric properties [8].

The genus *Ferula* L. (Apiaceae) is one of the most representative of richest in sesquiterpenes and their derivatives [9]. On the other hand, the genus *Ferula* is characterized by the presence of oleo-gum-resins (asafoetida, sagapenum, galbanum, and ammoniacum) and their use in natural and conventional pharmaceuticals [10]. Tissue adhesives derived from natural polymers are a more biocompatible alternative to synthetic glues for clinical medicine [11]. Kim *et al.* developed TAPE, a new class of hemostatic adhesive, which inspired by the water-resistant molecular interaction of a plant-derived polyphenolic compound with anti-oxidant, antibacterial, anti-mutagenic, and anti-carcinogenic properties [12].

What is interested in *Ferula ovina* is the small number of bioactive compounds including Ferutinin, Tschimgine and Stylosin that all of them are phytoestrogens [13, 14]. There is a growing body of literature that recognized osteoinductive potential of Ferutinin, one of the bioactive compounds of *F. ovina*, as a possible therapeutic for bone tissue engineering or for treatment against osteoporosis induced by estrogen deficiency.

Zare Mirakabad *et al.* demonstrated the positive effect of Ferutinin on bone mineralization of developing zebrafish larvae [15]. The compound could enhance bone reconstruction when orally administered in rats with a calvaria critical size bone defect, filled with collagen type1 and human amniotic fluid stem cells (hAFSCs). This construct leads to an approximately 70% bone reconstruction, which demonstrated Ferutinin could act as a healing promoting factor, on hAFSCs including osteogenic differentiation [16–18]. Zavatti *et al.* demonstrated Ferutinin is able to promotes proliferation and osteoblastic differentiation in human amniotic fluid stem cells and dental pulp-derived stem cells [17]. Rolph *et al.* indicated Ferutinin directs dental pulp-derived stem cells towards the osteogenic lineage by epigenetically regulating canonical Wnt signaling [19].

Dietrich *et al.* in comparison of skeletal feature in human and zebrafish indicated the bone type (dermal, compact and spongy), skeletal cell types (chondrocytes, osteoblast, osteoclasts and osteocytes) and ossification types (endochondral, Intramembranous and Perichondral) is similar in human and zebrafish [20]. On the other hand, skeletal development occurs in utero in human and extrauterine in zebrafish; bone mineralization in human begins after 4–5 weeks and reaches to maturity up to 30 years which starts in zebrafish at 3–4 dpf and matures after 2–4 months. In addition, cortical bone is not present in zebrafish and cartilaginous bones,

limited to the craniofacial skeleton and hematopoietic bone marrow is not bone-encapsulated in zebrafish [21, 22]. Therefore, zebrafish is a well-developed model system for studying both embryonic development and human diseases. It has also been known as an ideal *in vivo* model for the systematic identification of bioactive natural products with therapeutic capability and suitable model for screening of the agents that could prevent osteoporosis [23–27] and studying biocompatibility of dental materials [28–30]. Zebrafish embryos are commonly used to screen for toxicity and to study the effects of phytoestrogens [15, 31] on organ formation. Several studies also used zebrafish embryos to trace the effect of Estradiol on chondrogenesis and expression of related genes [31–33].

The first question in this research was finding the reason for observing four fractions in TLC of the root extract of *Ferula ovina*, while literature reviews reported just three compounds. With respect to the first research question, the starting point of the current study was the identification of Fenoferin as a novel terpenoid isolated from the root of *F. ovina*. Next, to quantify the effect of different phytoestrogens derived from *F. ovina* on craniofacial calcification, we measured the ceratohyal angle and calcified teeth and bone (ceratohyal) that could underlie the changes in craniofacial cartilage, bone (ceratohyal) and tooth mineralization in co-exposed zebrafish larvae relative to those exposed to either Ferutinin or Fenoferin alone.

## 2. Material and methods

### 2.1. Plant

**2.1.1. Plant material.** Whole plants of *Ferula ovina* were collected from Binaloud Mountain, Iran, in February 2017, and voucher specimens were deposited in the herbarium of Research Center of Plant Sciences, the Ferdowsi University of Mashhad under accession No. 24079-FUMH.

**2.1.2. Extraction, TLC and LC-MS analysis.** The plant material from *F. ovina* was extracted with dissolving powdered air-dried roots (50 g) in 150ml of 95% MeOH for 24 h at room temperature. The extracts were filtered (0.2 m filter) and dried separately in a rotary-evaporator (Buchi, Germany). Thin Layer Chromatography (petroleum ether-ethyl acetate as solvent) was optimized to compare fractions before performing column chromatography. According to the obtained fractions, the substances of concentrated extract of the root was separated by silica gel column (60×5 cm) chromatography (petroleum ether and subsequently ethyl acetate as solvents). Next, LC-MS carried out to analysis the extract and the last separated fraction according to Arnoldi *et al.* [34].

### 2.2. Zebrafish

**2.2.1. Zebrafish maintain, husbandry and embryo care.** All procedures involving zebrafish were performed in accordance with protocols approved by the University of Saskatchewan Committee on Animal Care and Supply and Animal Research Ethics Board (#200090108). Adult wild-type zebrafish maintained in an Aquatic Habitats Flow-Through System (Apopka, FL) on a 14/10 day/night cycle to mimic natural conditions and were fed with alive brine shrimp and chironomids (Hikari, Hayward, CA) at least once a day. The couples of wide type adult zebrafish (AB strain) mated in separated tanks, and the eggs were harvested in the next day. Healthy eggs which were recognized under microscope, were transferred into 0.5X E2 (7.5 mM NaCl, 0.25 mM KCl, 0.5 mM MgSO4, 75 mM KH2PO4, 25 mM Na2HPO4, 0.5 mM CaCl2, 0.35 mM NaHCO3, 0.5 mg/l Methylene Blue, pH ~ 7.0) and were incubated at 28˚C and dead embryos (opaque white rather than transparent) were removed. The remaining embryos were rinsed once more on the day of collection and every 24 hours thereafter. All embryos and larvae were kept in an incubator at 28˚C. In summary, the number of tanks

which were prepared for mating couples was 8 to 14 and in average 200 to 300 eggs were harvested from each couple if they laid eggs. All portions of the study have been performed on 3–5 clutches; each contains 100–150 healthy eggs.

**2.2.2. Embryo treatments.** Firstly, wild-type zebrafish larvae were exposed to Ferutinin (0.625, 1.25 and 2.5 μg/mL) and Fenoferin (0.05, 0.1, 0.5, 1, 5 and 10μg/ml) at 2 dpf, separately. Next, wild-type zebrafish larvae were exposed to combination of Ferutinin, Fenoferin, Stylosin and Tschimgine using the extract of the root of *F. ovina* (5 and 10μg/ml) and 0.1% concentration of DMSO, separately. The final DMSO concentration in embryo medium was 0.1% to avoid malformation and positive/negative false results [35, 36].

After plating treatments, zebrafish larvae at 2dpf were transferred into 24-well plates (3–4 larvae in 2ml EM per well) and incubated at 28˚C to assess effect of purified Ferutinin and Fenoferin compared to solvent and root extract which contains Ferutinin, Fenoferin, Stylosin and Tschimgine on craniofacial cartilage, bone (ceratohyal) and tooth mineralization.

## 2.3. Screening, staining and scoring

Zebrafish larvae were collected at 6dpf (when the cranial bone doesn't develop completely) and anesthetized with 0.4% tricaine (MS-222, Sigma). The anesthetized larvae of each clutch were placed into separated microtube to be fixed and stained based on two colors acid-free cartilage and bone staining protocol for zebrafish larvae [37]. Images of dorsal aspect head bone of zebrafish were then taken using a DFC310 FX camera (Leica, Germany) of M205 FA stereomicroscope (Leica, Germany).

To quantify the combinatorial effect, we measured the ceratohyal angle and calcified teeth and bone that could underlie the changes in craniofacial cartilage, ceratohyal and tooth mineralization.

In current study, the destination between Meckel's cartilage and ceratohyal defined as M-Ch, and the destination between Meckel's cartilage and notochord defines as head length, and the destination between two Opercles defined as head wide.

The first ratio related to the changes in craniofacial cartilage were calculated by dividing head wide to head length after measuring the length of head and head wide. The second and third ratio after measuring distance from Meckel's cartilage to ceratohyal revealed M-Ch to head length and M-Ch to head wide. We scored the changes in ceratohyal angle in four groups, including 0 for Ch-angle ≤ 70, 1 for 70 < Ch-angle < 90, 2 for Ch-angle ~ 90, and 3 for Ch-angle > 90.

We recognized tooth calcification based on the result of Karaman *et al.* [30]. Scoring the calcification performed based on red color to investigate alteration in bone mineralization. The scoring system with six scores for tooth included 1 for samples with just one tooth (4V1) on both posterior ceratobranchial arches, 2 for calcified 4V1 and 3V1, 2.5 for 4V1-3V1 and 5V1, 3 for 4V1, 3V1 and 5V1, 3.5 for 4V1, 3V1, 5V1, 4V2, and 4 for four and more calcified teeth, and six scores for ceratohyal included 0 for the target area without bone mineralization (red), 1 for bone mineralization of one side of target cartilage, 1.5 for bone mineralization of one side of target cartilage and two sides of paired cartilage, 2 for calcified two sides of cartilage but not completed, 2.5 for calcified two sides of cartilage with a bridge and the score 3 for completely mineralized bone (Fig 1).

## 2.4. Statistical analysis

Changes in craniofacial cartilage, bone (ceratohyal) and tooth mineralization were assessed for chemical concentration, which was tested in triplicate (number of larvae in each well) in at least three independent experiments (number of tested clutches). According to the number of mated couples (at least 3 clutches with 100–150 healthy eggs), the sample size of each treatment

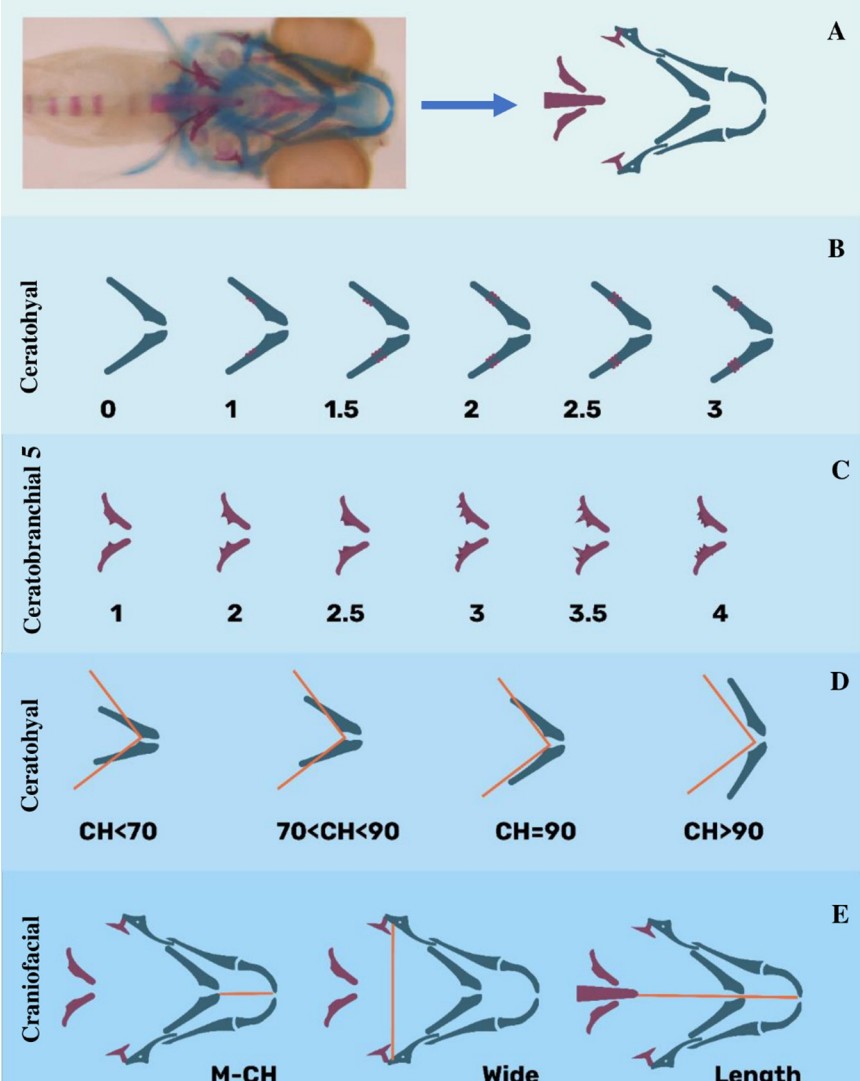

**Fig 1. Two color staining and scoring system of the bones, teeth and cartilages in treated zebrafish larvae at 6 dpf.**
(**A**) target craniofacial bones and cartilages related to a sample of the stained zebrafish larvae: blue parts represent ceratohyal and red parts demonstrate mineralized sections, (**B**) ceratohyal scoring system: 0 when the target area was just blue without bone mineralization, 1 for bone mineralization of one part of target cartilage (one row of osteoblasts), 1.5 for red stained one part of cartilage, but two parts of the paired one (one and two rows of osteoblasts), 2 for red stained two parts of cartilage (two rows of osteoblasts), 2.5 for red stained two parts of cartilage, but three part of the paired one (two and three rows of osteoblasts), and 3 was for completely mineralized bone(three rows of osteoblasts), (**C**) ceratobrancial 5 as location of teeth scoring system: 1 for samples with just one tooth (4V1) on both posterior ceratobranchial arches, 2 for calcified 4V1 and 3V1, 2.5 for 4V1-3V1 and 5V1, 3 for 4V1, 3V1 and 5V1, 3.5 for 4V1, 3V1, 5V1, 4V2, and 4 for four and more calcified teeth, (**D**) cartilage of ceratohyal angle scoring: 0 for Ch-angle $\leq$ 70, 1 for 70 < Ch-angle < 90, 2 for Ch-angle ~ 90, and 3 for Ch-angle > 90, (**E**) the destination between Meckel's cartilage and ceratohyal (M-Ch), the destination between two Opercles (head wide)the destination between Meckel's cartilage and notochord (head length) to calculate M-Ch: Wide, M-Ch: length, wide: length. Circles showed the calcified cells.

was at least 4. Data analyses were performed by using IBM SPSS Statics 19 as mean ± SEM and values were compared using one-way ANOVA followed by a Tukey's post-hoc pairwise test to compare bone mineralization of treatment concentration. For all statistical analysis, a P value <0.05 was considered to be significant. Concentration-response curves were modeled by the Microsoft Excel.

# 3. Results

## 3.1. LC-MS analysis of the last separated fraction by silica gel column chromatography

TLC results of concentrated methanol extract of root of *F. ovina* compared to Ferutinin (Sigma-Aldrich, Oakville, ON, Canada) indicated that Ferutinin appears as the first of four fractions [Fig 2D]. The results of LC-MS analysis of *Ferula ovina* extract compared to Ferutinin is depicted in (S1A and S1B Fig). The appeared signal 341 m/z in Q1 analysis $[M+H-H_2O]^+$ and the fragments of 121 m/z and 203 m/z in the $MS^+$ analysis of *Ferula ovina* extract confirmed that the root of *F. ovina* contains Ferutinin; but, the positive ESI-MS spectrum of unknown compound did not exhibit a signal at m/z 341 $[M+H-H2O]^+$ and a fragment at m/z 203 $[M+H-H_2O]^+$. Also, the negative APCI-MS spectrum did not show a signal at m/z 359 $[M-H]^+$. Therefore, the results of LC-MS analysis confirmed that it was not Ferutinin (S1 Fig), and NMR analysis as the complementary experiments identified it as Fenoferin (S2 Fig).

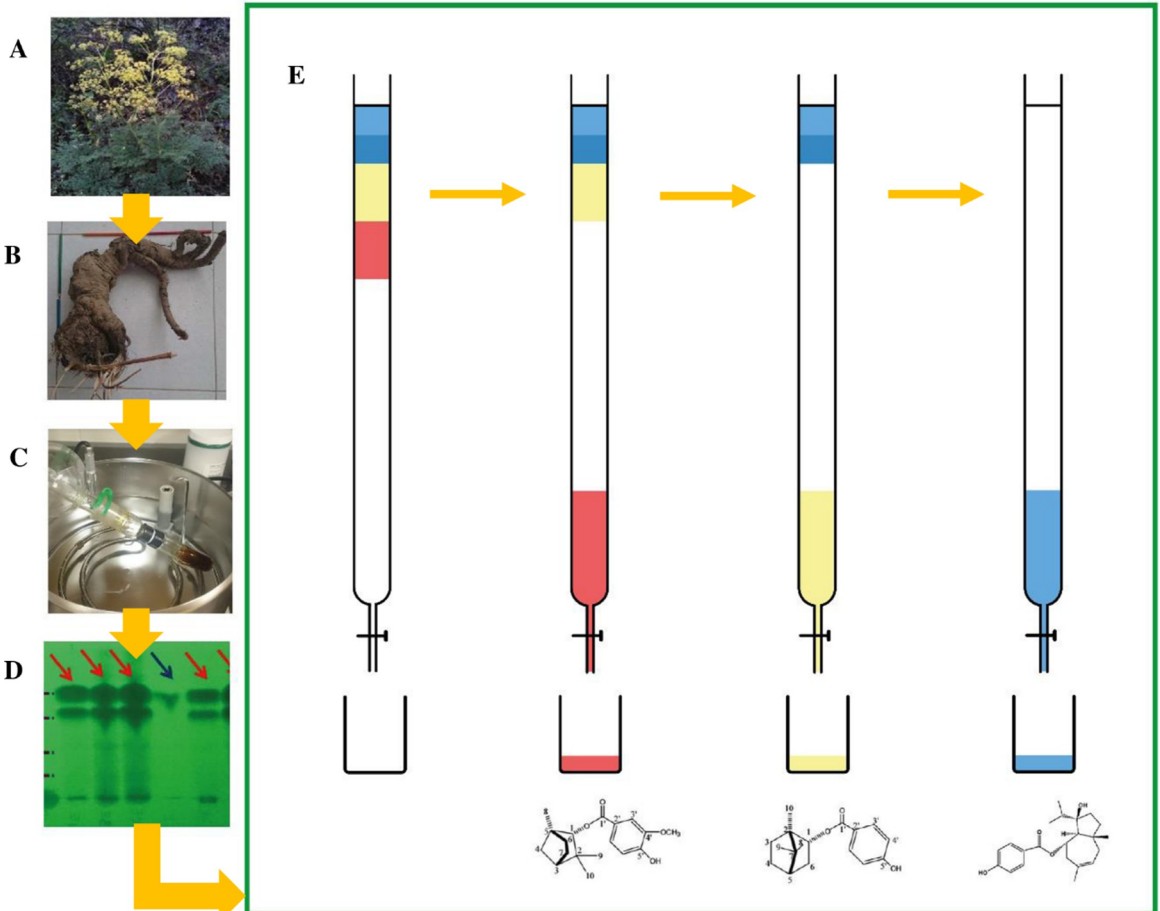

**Fig 2. The schematic process of column chromatography of the root extract of *Ferula. ovina*.** (A) Shoots of *F. ovina*, (B) Root of *F. ovina*, (C) extract of the root of *F. ovina*, (D) Thin Layer Chromatography (TLC) of methanol extract of the root of *F. ovina* compared to Ferutinin (Four fractions), Red allows for the result of extract samples, blue arrows for Ferutinin, purple dashed lines for the situation of fractions, (E) column chromatography of methanol extract of the root of *F. ovina*: red shows Stylosin, yellow represents Tschimgine, blue indicates Ferutinin.

### 3.2. Phenotypic readout of treated larval fish

We observed a range of phenotypes after staining with alcian blue-alizarin red (Fig 3). To compare the difference between treated samples, we defined four groups of changes for stained larvae: First group was the changes in craniofacial cartilage including the ratio which revealed changes in wide and length of the head; second one was for ceratohyal angle; third group was the changes in ceratohyal as represener of bone mineralization, and fourth group was the number and situation of teeth on ceratobranchial5. Fig 4 compares the intercorrelations among exposed zebrafish larvae to 12 different treatments.

**Craniofacial cartilage.** A positive correlation was found between three ratios. The highest ratio of M-Ch to length and M-Ch to wide was observed in the treated larvae with extract 5 which was significantly different in exposed larvae to Ferutinin (1.25 μg/ml) and Fenoferin (0.1 μg/ml). On the other hand, treated larvae with Ferutinin 1.25 indicated significant lowest ceratohyal angle. A negative correlation was found between three ratios and ceratohyal angle (Fig 4A and 4B).

**Ceratohyal mineralization.** The results demonstrated that ceratohyal mineralization occurred significantly in treated larvae with 1.25 μg/ml of Ferutinin and 0.1 μg/ml of Fenoferin (Fig 4C). The extract-treated larvae indicated the lowest amount of mineralization.

**Tooth formation.** The teeth 4V1, 3V1 and 5V1 grew in most of treated larvae, but 5V1 did not observe in the most of Extract-treated larvae while Fenoferin-treated larvae indicated the highest tooth formation, significantly (Fig 4D).

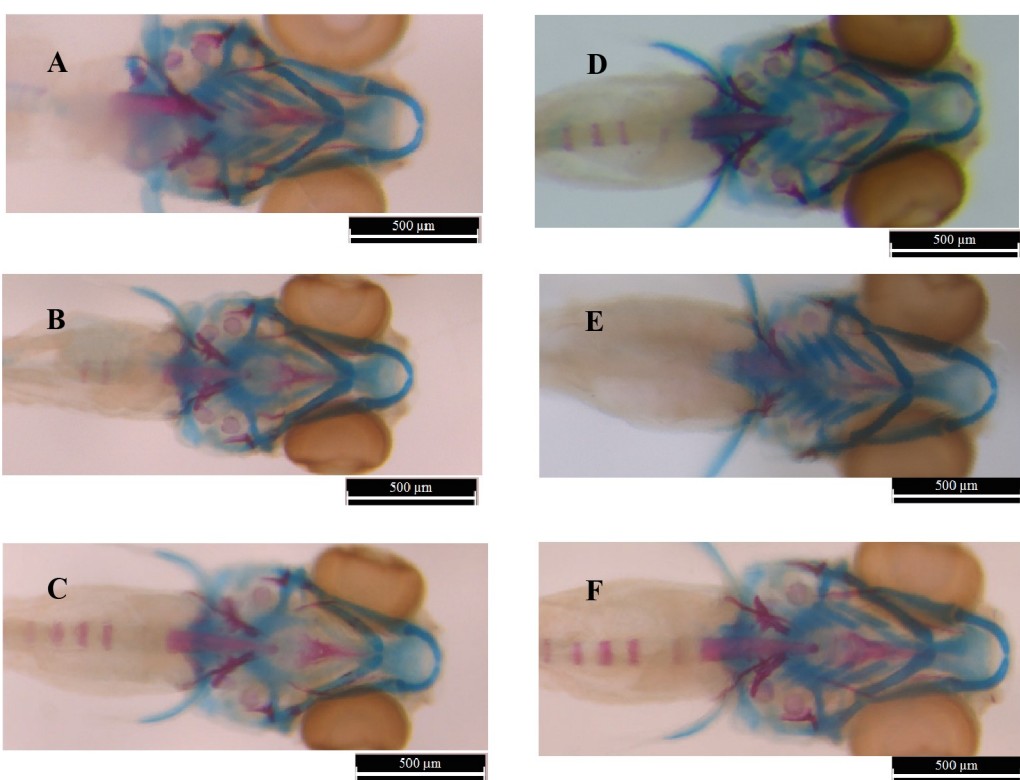

**Fig 3. The samples of alcian blue-alizarin red staining of the zebrafish larvae (6dpf) exposed to Fenoferin, Ferutinin and root extract of *Ferula ovina* compared to control.** Scale bars represent 500 μM. The blue parts represent ceratohyal and red parts indicate mineralized sections. (A) larvae exposed to 0.1 μg/ml of Fenoferin from 2dpf, (B) larvae exposed to 0.5 μg/ml of Fenoferin, (C) larvae exposed to 1 μg/ml of Fenoferin, (D) larvae exposed to 1.25 μg/ml of Ferutinin (E) larvae exposed to 5 μg/ml of root extract of *Ferula ovina*, (F) larvae exposed to negative control (DMSO).

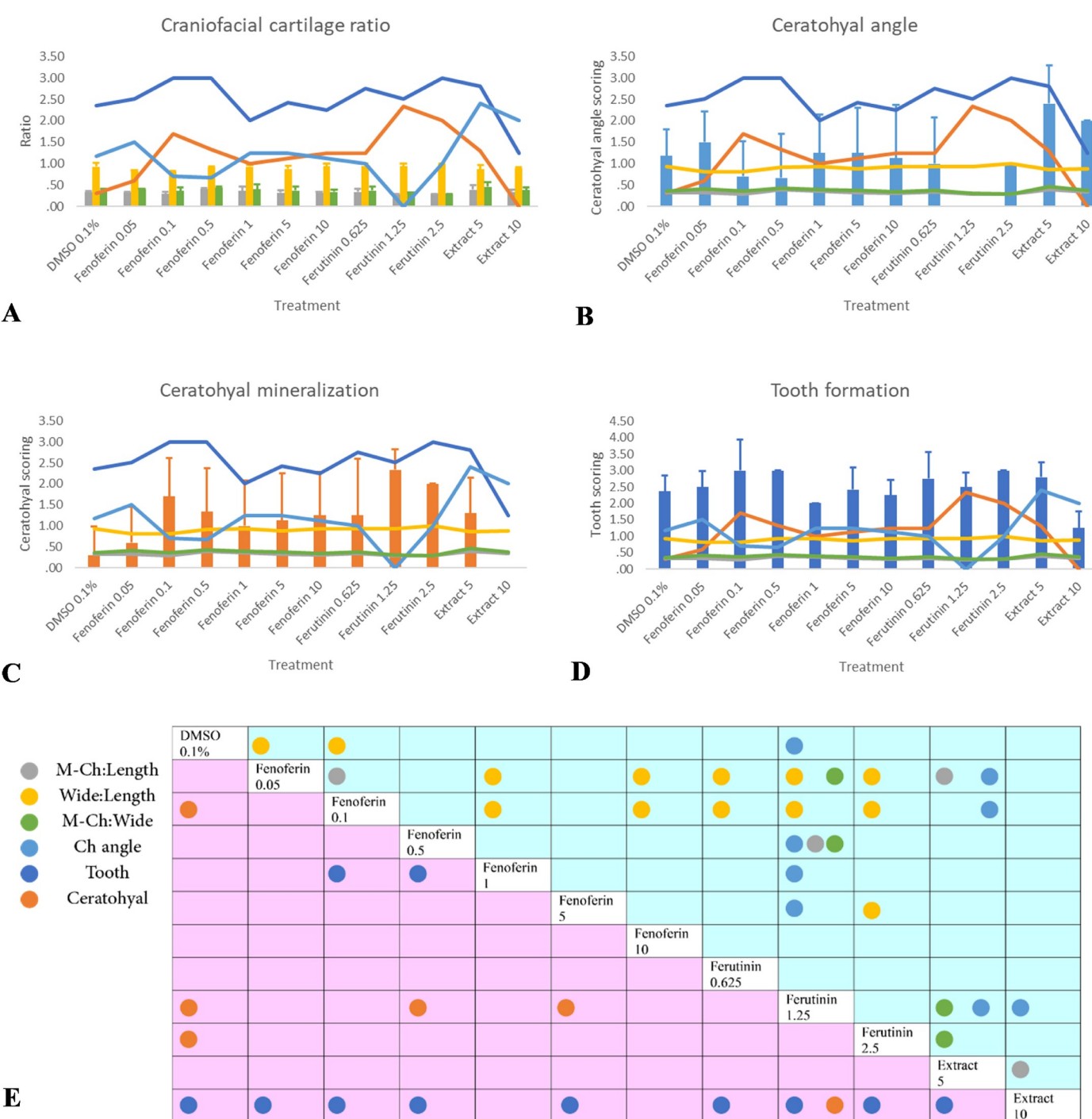

**Fig 4. Effect of concentration of Fenoferin, Ferutinin and root extract of *Ferula ovina* on bone mineralization, tooth formation and cartilage changes in treated zebrafish larvae.** Bars represent mean +SEM. Samples were analyzed in triplicate. (A) Craniofacial cartilage ratio: the gray bars represent M-Ch: Length; the yellow bars represent Wide: Length; the Green bars represent M-Ch: Wide, (B) Ceratohyal angle: the light blue bars indicate ceratohyal angle, (C) Ceratohyal mineralization: The orange bars represent ceratohyal mineralization (D) Tooth formation: The blue bars represent ceratohyal mineralization (E) The blue part represents cartilage changes including M-Ch: Length, Wide: Length, M-Ch: Wide and ceratohyal angle; the red part indicates bone mineralization and tooth formation. The circles represent groups with significant difference in mean, which were statistically different (* $p < 0.05$), by multiple comparison of means using one-way ANOVA and Tukey's post hoc test (gray circles for M-Ch: Length, yellow circles for Wide: Length, Green circles for M-Ch: Wide, light blue circles for ceratohyal angle, blue circles for tooth formation, orange circles for ceratohyal mineralization).

## 4. Discussion

### 4.1. Identification and characterization of Fenoferin in the root of *F. ovina*

We observed four fractions of TLC in methanol (current study) and dichloromethane [38] extract of the root of *F. ovina*, separately; but the findings did not support the result of last studies which reported just three compounds [Fig 2]. Therefore, an initial objective of the current paper was to identify an unknown compound.

Firstly, LC-MS analysis of methanol extract of the root of *F. ovina* was performed and confirmed Ferutinin as a sesquiterpene of the root. Up to now, far too little attention has been paid to third separated fraction because of overlapping with Ferutinin. LC-MS analysis of the third separated fraction compared to Ferutinin (Sigma-Aldrich, Oakville, ON, Canada) demonstrated that it was not purified Ferutinin, but the NMR analysis results identified it as Fenoferin.

Farhadi *et al.* suggested qHNMR instead of HPLC for quantification of Ferutinin because of the overlapping with other compounds [14], which confirmed the result of current study. Moreover, the polarity of Fenoferin is similar to Ferutinin and both of them get out from chromatography column at the same time which may affected ignoring it as the fourth bioactive compound and phytoestrogens derived from root of *F. ovina*.

### 4.2. Effect of the extract, Fenoferin and Ferutinin on bone mineralization

We designed the present study based on the result of Zare Mirakabad *et al.* [15] to determine the effect of phyroestrogens derived from *F. ovina* (Ferutinin, Fenoferin and root extract) on craniofacial cartilage, bone (ceratohyal) and tooth mineralization in exposed zebrafish larvae.

The result of current study showed significant increase in ceratohyal mineralization and significant decrease in ceratohyal angle in treated larvae with Ferutinin (1.25 μg/ml) which affected the ratio of MCH to wide, too; however, MCH to length and MCH to wide indicated similar range. Fenoferin-exposed larvae demonstrated significant changes in wide to length and MCH to length. Decreasing in tooth calcification observed in extract-treated larvae, significantly.

With respect to the research question, it was found that there is similarity between effect of Fenoferin in low concentration and Ferutinin in the ratios of head wide to head length, M-Ch to head length and M-Ch to head wide.

In comparison of the effect of Ferutinin [15] and Fenoferin, the significant bone mineralization was observed in the exposed larvae to 1.25 μg/ml of Ferutinin at 2dpf compared to positive and negative control while it was occurred in 0.1 μg/ml and 0.5 μg/ml Fenoferin-treated larvae compared to negative control. The result confirmed that *F. ovina* is one of the valuable plants with estrogenic root.

According to the result of current paper and the literature review, the root of *F. ovina* contains Fenoferin ($C_{17}H_{22}O_3$), a hydroxybenzoate with an aromatic ring and a hydroxyl group what is structurally similar to estradiol, Ferutinin ($C_{22}H_{30}O_4$), Tschimgine ($C_{17}H_{22}O_3$), and Stylosin ($C_{18}H_{24}O_4$) which have been known as phytoestrogen. The next question in this research was to quantify the effect of different phyroestrogens derived from *F. ovina* on craniofacial calcification in extract-exposed zebrafish larvae.

Since Fenoferin was predicted to mimic the effects of Ferutinin, we exposed wild-type zebrafish larvae to a combination of Ferutinin, Fenoferin, Stylosin and Tschimgine using the extract of the root of *F. ovina* (5 and 10 μg/ml) to examine effect of high concentration of combination of phytoestrogens on bone and tooth mineralization. To quantify this combinatorial effect, we measured the ceratohyal angle and calcified teeth and bone (ceratohyal) that could underlie the changes in craniofacial cartilage, bone (ceratohyal) and tooth mineralization and

observed a significant reduction in co-exposed larvae relative to those exposed to either Ferutinin or Fenoferin alone, suggesting a strong competitive interaction (Fig 4B).

Gliozzi *et al.* [39] and Macrì *et al.* [8] inferred to protective potential of sesquiterpenes at low doses and severe toxic effects at high doses is due to inducing calcium ions by sesquiterpenes and increasing cation permeability of lipid bilayer and mitochondrial membrane in a dose-dependent manner with a higher selectivity for divalent cations. We observed negative effect of high concentration of phytoestrogens on bone mineralization and tooth formation. Comparison of the findings with those of other studies confirms toxic effects of root extract at high doses and protective potential of Ferutinin and Fenoferin at low doses.

As mentioned in the literature review, Ferutinin at low-concentration is associated with an important antioxidant action and promotes proliferation and osteoblastic differentiation in human amniotic fluid stem cells and dental pulp-derived stem cells through the regulation of Wnt/catenin pathway [8, 17, 19]. These results provide further support for the hypothesis that root extract of *F. ovina* with oleo-gum-resin and their use in natural and conventional pharmaceuticals is a suitable candidate for regeneration medicine and tooth implant research.

To develop a full picture of impact of the root extract on dental implant due to its estrogenic bioactive compounds and adhesive properties, additional studies would be needed such as using it in mixture with ceramic biomaterial and used as composites to provide better integration of tissue for dentistry application.

## Supporting information

**S1 Fig.** Mass Spectrometric (MS) analysis of Ferutinin, Ferula ovina extract and related unknown fraction of TLC: (A) LC-APCI(+) scan of Ferutinin standard (Ferutinin major ion m/z 341.2 = [M+H-H2O]+), (B) MS/MS of Ferutinin Standard (341.2: 203 and 121), (C) MS/MS Ferutinin (Direct infusion of F. ovina extract), (D) LC-MS/MS spectrum of F. ovina extract containing Ferutinin (MRM transitions 341.2: 203.2 represented by blue in spectrum, 121.1 represented by red in spectrum), (E) LC-ESI (+) scan of the unknown fraction of TLC did not present Ferutinin major ion m/z 341.2 = [M+H-H2O]+, (F) LC-ESI (-) scan of the unknown fraction of TLC did not present spectrum as Ferutinin, (G) LC-APCI(+)-MS/MS spectrum of the unknown fraction of TLC, (H) LC-ESI(+)-MS/MS spectrum of the unknown fraction of TLC.
(DOCX)

**S2 Fig. NMR of an unknown compound isolated from root extract of *Ferula ovina*.**
(DOCX)

**S3 Fig.**
(PNG)

**S1 Checklist.** *PLOS ONE* **humane endpoints checklist.**
(DOCX)

**S1 File.**
(PDF)

**S1 Graphical abstract.**
(DOCX)

## Acknowledgments

We should thank Brian F. Eames, Ed Krol and Anas El-Aneed for their advice and lab facilities at University of Saskatchewan (department of Anatomy, Physiology and Pharmacology).

## Author Contributions

**Conceptualization:** Hoda Zare Mirakabad.

**Data curation:** Hoda Zare Mirakabad.

**Formal analysis:** Hoda Zare Mirakabad.

**Investigation:** Hoda Zare Mirakabad.

**Methodology:** Hoda Zare Mirakabad.

**Project administration:** Hoda Zare Mirakabad.

**Resources:** Hoda Zare Mirakabad.

**Software:** Hoda Zare Mirakabad.

**Visualization:** Hoda Zare Mirakabad.

**Writing – original draft:** Hoda Zare Mirakabad.

**Writing – review & editing:** Hoda Zare Mirakabad, M. Reza Khorramizadeh.

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
