## [Decision Letter · Decision Letter 0]

16 Dec 2021

Introduction to the potential of Ferula ovina in dental implant research due to estrogenic bioactive compounds and adhesive properties

PONE-D-21-29536

Dear Authors,

We’re pleased to inform you that your manuscript has been judged scientifically suitable for publication and will be formally accepted for publication once it meets all outstanding technical requirements.

Kind regards,

Kelvin Ian Afrashtehfar, M.Sc., D.D.S.,Dr. med. dent., FRCDC

Academic Editor

PLOS ONE

Additional Editor Comments:

Dear Authors,

Congratulations for the acceptance of your manuscript in Plos One.

Kind Regards,

The Royal Academic Editor

Reviewers' comments:

Reviewer's Responses to Questions

**Comments to the Author**

1. Is the manuscript technically sound, and do the data support the conclusions?

Reviewer #1: Yes

2. Has the statistical analysis been performed appropriately and rigorously? 

Reviewer #1: Yes

3. Have the authors made all data underlying the findings in their manuscript fully available?

Reviewer #1: Yes

4. Is the manuscript presented in an intelligible fashion and written in standard English?

Reviewer #1: Yes

5. Review Comments to the Author

Reviewer #1: The manuscript was written with good English language. The introduction represents the importance of this study. Moreover, the methodology achieve all requirements to reach the goal of this study. The results were represented good with tablesand figures. The discussion appears sufficiently with conclusion.

6. PLOS authors have the option to publish the peer review history of their article (what does this mean?). If published, this will include your full peer review and any attached files.

Reviewer #1: No

---

## [Editor Report · Acceptance letter]

26 Dec 2021

PONE-D-21-29536 

Introduction to the potential of *Ferula ovina* in dental implant research due to estrogenic bioactive compounds and adhesive properties 

Dear Dr. Zare Mirakabad:

I'm pleased to inform you that your manuscript has been deemed suitable for publication in PLOS ONE. Congratulations! Your manuscript is now with our production department. 

Kind regards, 

on behalf of

Dr. Kelvin Ian Afrashtehfar 

Academic Editor

PLOS ONE